# Patient data-sharing for immigration enforcement: a qualitative study of healthcare providers in England

Vasiliki Papageorgiou ,[1] Alexandra Wharton-Smith,[2,3] Ines Campos-Matos,[3,4] Helen Ward[1]

[1]Patient Experience Research Centre, Imperial College London School of Public Health, London, UK
[2]Department of Global Health and Development, London School of Hygiene and Tropical Medicine Faculty of Public Health and Policy, London, UK
[3]Migration Health, Health Improvement Directorate, Public Health England, London, UK
[4]Collaborative Centre for Inclusion Health, University College London Institute of Epidemiology and Health Care, London, UK

**Correspondence to**
Vasiliki Papageorgiou;
vasiliki.papageorgiou17@imperial.ac.uk

## ABSTRACT

**Aim** To explore healthcare providers' perceptions and experiences of the implications of a patient data-sharing agreement between National Health Service (NHS) Digital and the Home Office on access to NHS services and quality of care received by migrant patients in England.

**Design** A qualitative study using semi-structured interviews, thematic analysis and constant-comparison approach.

**Participants** Eleven healthcare providers and one non-clinical volunteer working in community or hospital-based settings who had experience of migrants accessing NHS England services. Interviews were carried out in 2018.

**Setting** England.

**Results** Awareness and understanding of the patient data-sharing agreement varied among participants, who associated this with a perceived lack of transparency by the government. Participants provided insight into how they thought the data-sharing agreement was negatively influencing migrants' health-seeking behaviour, their relationship with clinicians and the safety and quality of their care. They referred to the policy as a challenge to their core ethical principles, explicitly patient confidentiality and trust, which varied depending on their clinical specialty.

**Conclusions** A perceived lack of transparency during the policy development process can result in suspicion or mistrust towards government among the health workforce, patients and public, which is underpinned by a notion of power or control. The patient data-sharing agreement was considered a threat to some of the core principles of the NHS and its implementation as adversely affecting healthcare access and patient safety. Future policy development should involve a range of stakeholders including civil society, healthcare professionals and ethicists, and include more meaningful assessments of the impact on healthcare and public health.

## INTRODUCTION

In 2017, a memorandum of understanding (referred to here as the data-sharing agreement) was agreed between the UK Home Office and National Health Service (NHS) Digital, with the Department of Health and Social Care acting as a signatory.[1] This formalised an existing arrangement, in place since 2013, whereby non-clinical patient data

### Strengths and limitations of this study

► To our knowledge, this is the first exploratory study of the experiences and perspectives of healthcare providers in relation to the patient data-sharing agreement between National Health Service Digital and the Home Office.
► The interviews generated rich, contextual data.
► The findings were interpreted using a social constructionist approach, acknowledging the researcher's own biases through a reflexive process and discussed with others to provide additional rigour.
► The main study limitation was sample size due to the limited availability of healthcare providers and difficulties with recruitment.
► Participants were also aware of the study purpose prior to taking part which may have influenced recruitment and the findings.

(online supplementary 1) were being shared to trace suspected immigration offenders, without patient consent nor sometimes the knowledge of clinicians; in some cases, primary care practices were asked directly by the Home Office to confirm patients' addresses.[1 2] NHS Digital ruled the process as justified as the information requested excluded clinical data and 'lies at the least intrusive end of the spectrum of confidential information'.[3] However, concerns were raised by the public health community (including researchers and clinicians) and advocacy groups, about the potential impact of the data-sharing agreement on access to NHS services by diverse and often vulnerable migrant populations, and to the wider public health, including sexual and reproductive health.[4–7] Following these objections, an amendment was made by the UK government (in the House of Commons) on 9 May 2018, to restrict requests to individuals being considered for deportation and convicted of 'serious' criminal offences, or presenting a risk to public security, to ensure compliance with the newly introduced EU General Data

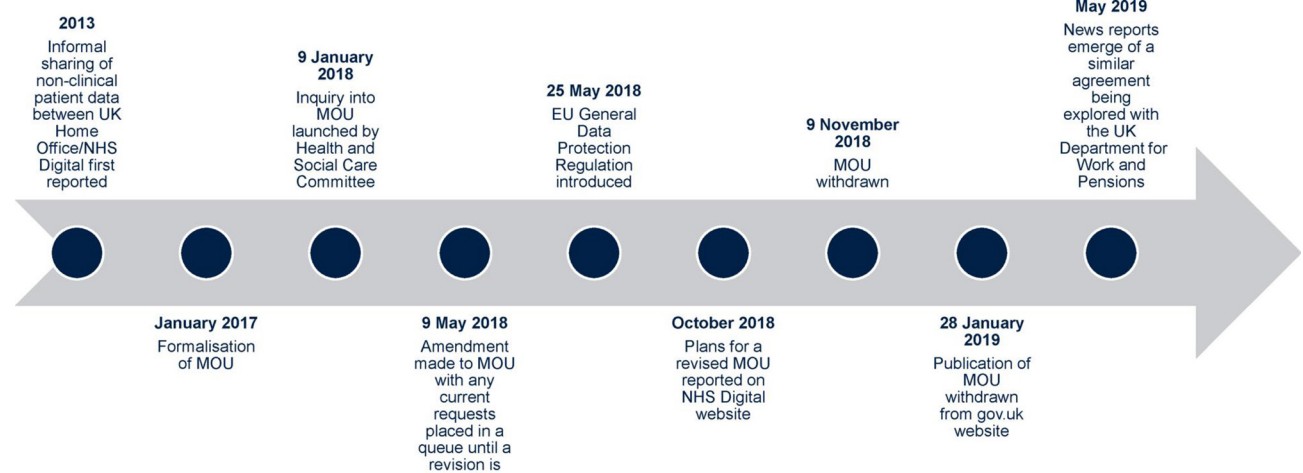

**Figure 1** A timeline of the memorandum of understanding (MOU) between the UK Home Office/NHS Digital formalising a sharing of non-clinical patient data for immigration enforcement purposes. EU, European Union; NHS, National Health Service.

Protection Regulation (GDPR) on 25 May 2018.[8–12] GDPR was introduced by the European Parliament and Council to provide greater rights and increased transparency to all European Union/European Economic Area (EU/EEA) citizens in the control, management and processing of their 'personal data' by organisations based within, and outside of, the EU.[11 12] The new UK Home Office/NHS Digital data-sharing agreement has yet to be published. However, a revision plan has been detailed by NHS Digital and the original agreement has now been withdrawn.[1 13] Recently, proposals for a similar agreement to determine health-related welfare entitlements between the NHS and Department for Work and Pensions have also been reported.[14 15] A summary of the historical context to the patient data-sharing agreement is given in figure 1.

The decision to migrate is often influenced by both internal and external factors; for instance, perceived prosperity leading to voluntary migration or an increased risk of violence due to conflict resulting in forced migration.[16–19] As a result, the diverse migration journey, including life before and after settling in a new place of residence, can have both a negative and positive influence on health and well-being outcomes, in the short term, medium term and long term.[16 20] In addition, various factors have been shown to influence the health-seeking behaviour and access to healthcare services of documented and undocumented migrants, including cultural ideas and expectations, socioeconomic conditions, geographical and institutional factors, the extent and nature of which may vary according to contexts.[5 16 21] However, important gaps exist in the literature on migrant health, health-seeking (or avoidance) behaviour and access generally; national and international comparisons are challenging due to differing definitions, contexts and limited epidemiological data.[22 23] For instance, research conducted in Spain has found both a higher and lower utilisation of general practitioner (GP) services by different groups of migrants, compared with the native population, due to inequities when seeking initial contact.[24–26] Studies globally, including the USA, have demonstrated that perceived fear of deportation may act as a barrier to migrants accessing health services and has been associated with poor health outcomes including mental health.[27–31] To date, most studies exploring healthcare registration, access or utilisation of services by migrants focus on specific subgroups, populations and areas within a country and are either qualitative[32] or observational studies (including cross-sectional or a retrospective cohort).[23 33 34] Their limitations include, findings not being generalisable to other settings, or not reflecting the transient nature of migration as they only provide a 'snapshot' in time or being influenced by attrition bias.

Within England, Britz and McKee[5] have described how healthcare and living condition inequities can result in preventable morbidity and mortality of vulnerable groups, including migrants, resulting in higher healthcare costs. For instance, charging regulations in England are used to determine eligibility for free NHS services with these guidelines being reported as a barrier for migrants accessing healthcare.[5 35 36] Even where services are free, for example, in primary care, access may be limited by registration processes and practices which can vary by GP practice.[37] Additionally, documented and undocumented migrants are often unaware of their entitlements and report mixed experiences during registration, including language barriers.[32] Consequently, there are concerns that unregistered individuals are shifting costs to secondary care due to late presentation with greater clinical severity.[38 39] Discontinuity in care, felt hostility or perceived stigma can further increase difficulties while navigating the health system; the healthcare system itself may act as a social determinant.[40 41]

The broader, international, political context has begun to shift towards stricter immigration and border control; for instance, restrictive border control introduced by the Trump administration in the USA[42] and limited entitlements to healthcare by undocumented migrants accessing healthcare services across areas of Europe

(eg, Finland and Ireland).[21 43] Additionally, concerns have been reported about the collection of individuals' migration status in healthcare systems globally, particularly regarding the potential misuse of health and identifiable data, including immigration status and address.[16 29] Critics emphasise that patient confidentiality and trust are incompatible with the UK Home Office/NHS Digital data-sharing agreement and NHS charging regulations.[5 10 16 29 44 45] These concerns reflect broader political contexts; a 'hostile environment' has been explicitly promoted by the UK government in relation to migration, evident in the application of immigration policy to areas including social welfare since 2010.[46 47] For instance, the UK Immigration Act 2016, explicitly describes measures to restrict support given to individuals (and their dependants) with a rejected asylum seeker application.[48 49] This sits uncomfortably with the increased onus on public and organisational bodies to regulate health and social care data following the introduction of GDPR.[12] Specifically, the UK's independent authority (Information Commissioner's Office) and network of UK Caldicott Guardians are responsible for managing data access requests and protecting an individual's confidentiality within NHS organisations and local authorities providing social services.[11 50 51] Additionally, there is limited research, in both a global and UK context, on the potential conflicts between these different parts of government; for example, on the impact of sharing healthcare data for the specific purpose of immigration enforcement.[44–46 50 52–54]

Therefore, it is important to understand the perceptions of the UK Home Office/NHS Digital data-sharing agreement by UK healthcare workers (including those in primary, secondary and tertiary care) as migrants may access healthcare at each of these services. This study aims to contribute to the growing evidence of predominantly grey literature,[53] perspectives[46] and patient case studies,[44 45 54] by providing robust data on the experiences of the data-sharing agreement among healthcare providers and voluntary sector workers in England. Our aim was to explore whether the policy has influenced clinical care provided and healthcare received by migrant patients and possibly decision-making processes in clinical care and healthcare data management, particularly as a new agreement is currently being drafted. To our knowledge, no previous primary research has been published focusing on this topic among UK healthcare workers.

## METHODS
### Study design
We carried out a qualitative study using interviews with healthcare providers and a voluntary sector worker. The study was underpinned by a social constructionist approach to explore the connection, power and inequality between the 'macro' (eg, public health), 'meso' (eg, Home Office and NHS Digital) and 'micro' levels (eg, clinical practice), which were contextualised by individual's responses and recognised the interviewer's

own biases.[55 56] We drew on the NHS principles of the NHS Constitution for England as a framework, as previously performed by Rafighi et al,[57] to inform interviewees' perceptions and experience of migrants' access to health services and migrants' health-seeking behaviour.

### Recruitment and sampling
Participants were 11 healthcare providers (10 medical doctors, 1 physiotherapist) and 1 non-clinical volunteer in urban areas of England with experience supporting migrants accessing healthcare services.

Initially, our sampling strategy was to recruit healthcare providers based at NHS sites with large migrant populations (eg, London). We collected broad geographical location data from study participants and so we are unable to comment on the level of migration at participant locations. To comply with University ethics, we were unable to directly recruit individuals from NHS sites; therefore, participants were recruited from professional bodies, professional networks, social media and existing personal contacts (VP, HW, IC-M and AW-S) by written email invitation.

Selection initially focused on primary healthcare providers and voluntary sector workers, but we expanded to include healthcare professionals in secondary care following slow initial recruitment due to unavailability and delayed response.

Participants were sent an information sheet and consent form prior to the interview taking place and given the opportunity to ask any questions about the study. All participants gave written informed consent.

### Data collection
A member of the research team (VP) conducted semi-structured, in-depth interviews in July 2018, either in person (n=7) at the participant's workplace (non-NHS Trust site) or the university campus or by telephone/online (n=5). Recruitment was stopped at 12 participants due to time constraints of the study. Despite this, we began to see repetition of themes during the interviews.

An initial topic guide was used to encourage participants to describe their experiences and views of migrants accessing healthcare, both generally and in relation to the data-sharing agreement. The interview focused on the 'how', rather than the 'what', in line with the constructionist concept of this interaction.[58] Some themes were determined a priori, with emergent themes explored through subsequent interviews and the relationships between concepts investigated. Four initial themes were explored in the topic guide (online supplementary 2) and pilot tested with HW and AW-S. If participants were unaware or indicated little understanding of the patient data-sharing agreement, a short description was given. Probes were used to encourage participants to elaborate their answers and descriptive field notes were recorded by VP. All interviews were audio recorded and transcribed verbatim by VP.

## Data analysis

Memos were written on transcripts by VP describing initial reflections, with the findings analysed using an inductive–deductive approach through constant comparison, derived from grounded theory.[55][59] QSR International's NVivo 12 software was used to organise the data, with a primary thematic framework provided by the topic guide. Coding was completed by VP through initial line-by-line content analysis with subsequent focused coding and constant comparison between codes to generate emergent themes and subthemes.[55] HW and AW-S provided consensus on coding by checking the coding framework of the first three transcripts and were approached by VP to check any interpretations made on subsequent transcripts.

A thematic analysis, featuring summarised NHS principles (online supplementary 3), was used to organise data, where appropriate. The framework was not applied to directly categorise data as this would restrict interpretation and overlook the philosophical and empirical concepts being determined.[60]

The findings were shared with Public Health England to contribute to their review[4] commissioned by the Department of Health and Social Care. They were also shared with all participants as a lay summary by email. The standards for reporting qualitative research checklist has been used to report study methods and findings (online supplementary 4).[61]

## Patient and public involvement

There were no funds or time allocated for public involvement due to the timing of this postgraduate research project.

# RESULTS

## Study population/characteristics

Twelve participants were interviewed, with most based in London (table 1). Interviews lasted, on average, for 40 min (range 20–60). Participants worked in a range of specialties: general practice (n=4), infectious diseases and/or sexual health (n=4), psychiatry (n=1), physiotherapy (n=1), paediatrics (n=1) and community engagement (n=1). Interviewees included an equal split of men and women and a range of ages and levels of experience.

Interviewees' knowledge about the data-sharing agreement and their understanding varied (four provided a full description, five little description and three provided no description). Overall, most participants were confused about the specifics of the policy. Participants expressing some understanding often alluded to having learnt about the agreement either through the media or from colleagues making any subsequent answers an immediate reflection, rather than a considered opinion.

Views on the actual and potential impact of the agreement were embedded in a broader understanding of the challenges facing migrants accessing healthcare and are described here as three themes (figure 2): (1) patient experience, (2) ethical principles of public health and (3) power and relationships. These encompassed the themes explored in the topic guide. The themes interconnected with perceived consequences of the agreement and followed the natural sequence of the patient pathway from accessing to receiving healthcare in the NHS. This included reflections on any changes to clinical practice delivered and the patient–healthcare professional relationship.

**Table 1** Summary of study participants' demographics

| Participant Identifier | Age (years) | Gender | Clinical setting* | Country of birth | Location of workplace |
|---|---|---|---|---|---|
| B01 | 25–34 | F | Community | UK | London |
| B02 | 45–54 | F | Community | UK | Other |
| B03 | 35–44 | M | Hospital | Non-UK | London |
| B04 | 25–34 | M | Hospital | UK | London |
| B05 | 45–54 | M | Hospital | UK | London |
| B06 | 25–34 | M | Hospital | Non-UK | Other |
| B07 | 35–44 | M | Community | UK | London |
| B08 | 55–64 | M | Community | UK | London |
| B09 | 45–54 | F | Community | Non-UK | London |
| B10 | 35–44 | F | Community | Non-UK | London |
| B11† | 65+ | F | N/A—voluntary sector | – | – |
| B12 | 25–34 | F | Hospital | UK | London |

*Community (self-referral, primary care and sexual health services) or hospital settings.
†Incomplete data received.
F, female; M, male; N/A, not applicable.

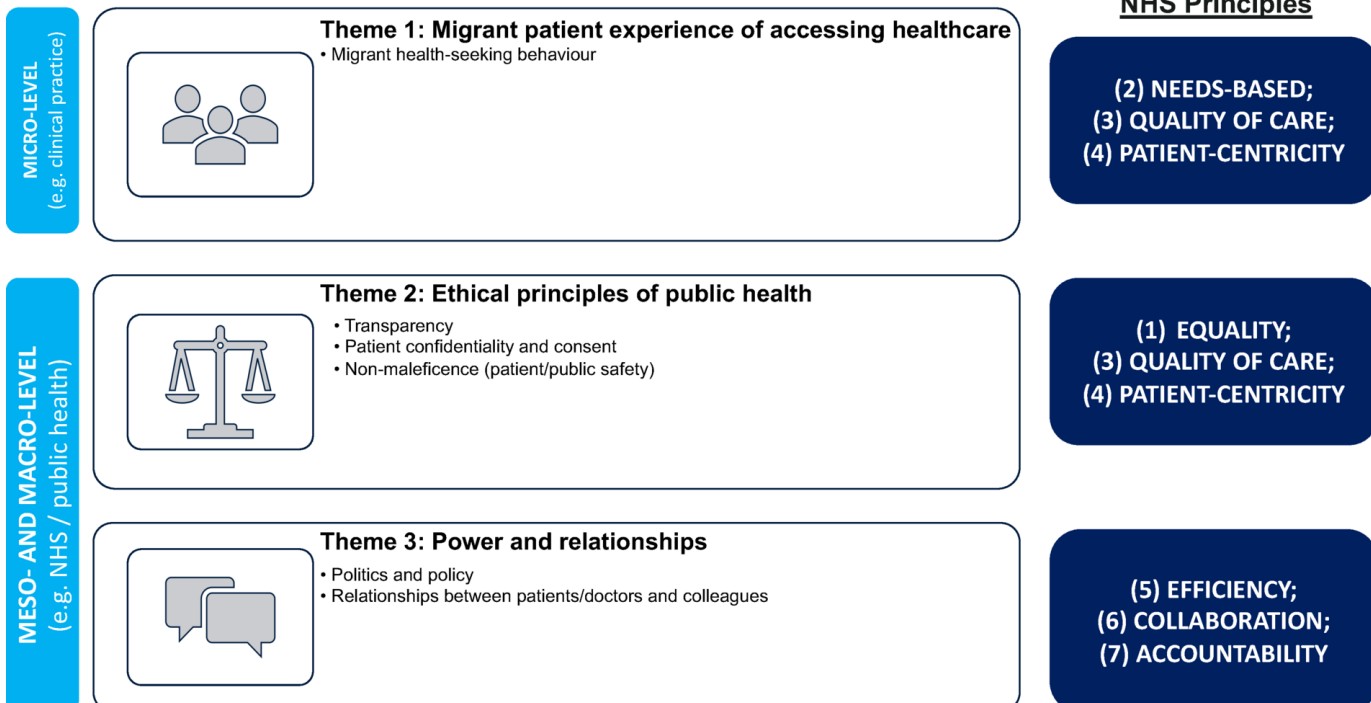

**Figure 2** A conceptual thematic framework of key themes and subthemes identified through data analysis used to organise findings. The framework also explores each theme's relationship with the NHS principles and at an individual (micro), organisational (meso) and societal (macro) level. NHS, National Health Service.

## Theme 1: migrant patient experience of accessing healthcare

Healthcare providers described their interactions with migrant patients along the care pathway, although confusion about the asylum process, healthcare entitlements and definition of the word 'migrant' were often described, either as a reflection of their own, or colleagues', practice.

### Migrant health-seeking behaviour

Participants described a range of influences on migrants' health-seeking behaviour, with some influences relating directly to enforcement and data-sharing and others concerning broader aspects. More than half of participants described discriminatory practices in relation to proof of eligibility:

> …informally we sort of found out the people with 'foreign-sounding' names were being targeted for these letters [from NHS Spine portal to confirm their home address if they had not used an NHS service in 6 months]. [B02]

This linked to suggestions that migrants would experience fear and anxiety around registration and accessing health services. Specific references were made to immigration control and patients' fears of arrest or deportation of either themselves or others in their immediate networks by the Home Office:

> …I think there is a group of people that are afraid and think that by coming into contact with services, they could […] put themselves on the radar. [B05]

Participants talked about how these resultant fears of accessing health services could be particularly harmful for vulnerable people, for example, those affected by mental illness. This reflects the backgrounds and interests of some interviewees; one described a suicidal UK citizen detained under the Mental Health Act whose husband was at risk of deportation and had received a letter from the Home Office:

> …we continuously sort of had to reaffirm the limits of the information that we share and how much of it is strictly confidential…. [B06]

Other perceived influences on health-seeking behaviour included: attitudes to health, cultural ideas and expectations of healthcare services. Participants described how these may result in either delayed presentation or the overuse of healthcare, which could influence public health funding and access to healthcare. For instance, participants working in sexual health and infectious disease services described how delayed or deferred clinical presentation could shift expenses from primary to secondary care:

> …if you delay somebody coming forward for HIV care, they're more likely to end up needing an inpatient admission which is more expensive… [B07]

Figure 3 shows the patient pathway and potential barriers migrants may face when accessing healthcare, which are also influenced by wider determinants. These were identified from the literature and further explored,

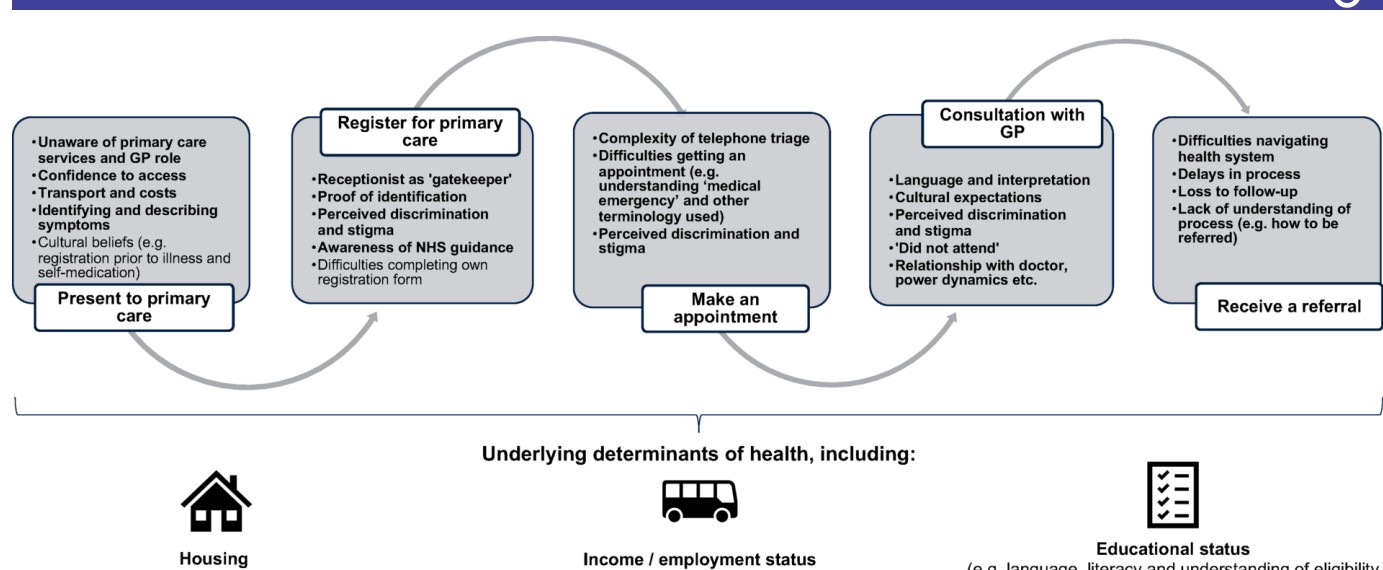

**Figure 3** The patient pathway for primary care services and the perceived barriers migrant patients may face as described by participants and the literature. Perceived barriers highlighted in bold were described during interviews. GP, General Practitioner; NHS, National Health Service.

in relation to the data-sharing agreement, during the interviews.

## Theme 2: ethical principles of public health
### Transparency
Most participants described migrant patients and colleagues as being unaware and having a limited understanding of the policy. They linked this to a 'sneaky' agenda, or evasion of transparency by the Home Office to access data without patients' nor clinicians' knowledge:

> …GPs have no idea that this precious data they hold is being accessed, sort of on the sly, without them even realising… [B01]

Concerns were also raised about how patients first hear about the agreement which could generate fear. A non-clinical specialist working for a community-based organisation supporting marginalised populations offered insight into knowledge exchange by rumours, which may influence clinical presentation:

> …they will go and share that with all their friends, family […] and they will panic. [B11]

### Patient confidentiality and consent
Participants felt that the data-sharing agreement directly contradicted the ethical principles of confidentiality and consent that underpinned the professional guidelines they abided by, resulting in an 'erosion in trust in healthcare' [B03]. Specifically, participants working in sexual health described the importance of ensuring confidentiality to individuals accessing these services whereas the participant working in mental health services described a difficult balance between managing 'risk' and confidentiality.

One question asked for participants' opinion of the policy in relation to patient confidentiality; four described practices they had taken or were prepared to take to protect confidentiality, which was in line with advice given by Doctors of the World[62] such as changes in recording patient data by keeping separate notes, or writing in shorthand:

> …then you'd be thinking […] 'I better not put that [telephone] number in the actual [notes], I'm gonna keep that'…Maybe it's over the top but it does create this kind of space for suspicion. [B04]

### Non-maleficence (patient and public safety)
Some participants suggested how migrant patients may bypass the agreement by giving false information or pretending to be someone else, which could impact both individual patient safety and that of the wider public. B04 describes a case shared by a colleague where this had occurred, and its implications:

> …there was a pregnant woman who came to deliver her baby and she was so worried about it she […] used her sister's details… The time that they knew that, was the time they came to give her a blood transfusion, and they realised that her blood group was different… [B04]

The concept of safety was further explained from the perspective of health protection and public security. Participants reflected on the public safety rationale of the policy, following the recent amendment, offering differing views, with some being less definitive:

> From an infectious disease, public health prevention point of view, 'are the people that are likely to have like transmissible illnesses more likely to be the type

of people that have committed serious offences?'
No… [B03]

One hospital-based participant spoke of 'security' in terms of the threat of terrorism, from the perspective of having witnessed a terrorist attack, and appeared unsure,

…I think as far as security goes… I don't know is the answer. [B12]

Participants were able to appreciate 'exceptional circumstances' [B02] where confidentiality may be breached, including criminal investigations or where patients may be at risk of harm to themselves or others, which some described as a 'rarity' in their own experience. However, most participants disagreed with the use of the agreement for immigration purposes, often referring to the 'philosophy' or principles of the NHS or medical profession.

### Theme 3: power and relationships
#### Politics and policy
Participants described the political climate, without probing, referring variously to the 'hostile environment', Brexit, healthcare charges or other policies they viewed as deterring migrants from accessing healthcare. One community clinician described migration as a political issue associated with 'scaremongering' by the media, and some participants suggested the need for governmental accountability and transparency. They proposed conducting an informed public debate on the agreement, indicating a lack of awareness of the 'open' status of the Health and Social Care Committee inquiry.[63]

Most interviewees mentioned a loss of faith in governmental decision-making processes, specifically the Home Office, following recent media attention on data-sharing and immigration scandals, predominantly Windrush. The 2018 Windrush scandal refers to the generation of Caribbean-born British citizens who entered the UK over four decades ago, some of whom were wrongly deported and denied re-entry.[16 64] Some participants also raised concerns of the potential misuse of the agreement by other government agencies, for instance, during benefit fraud investigations.

#### Relationships between patient/healthcare professionals and colleagues
Participants described trust, empathy and rapport as essential elements to the patient–healthcare professional relationship, which they perceived to be challenged by the agreement. Their narrative of the importance of this relationship were described with patients generally:

…if you've got, going to have a working relationship with somebody about the most intimate details of their lives […] you have to have that trust. [B02]

Participants described the positions of other colleagues on the data-sharing agreement. One expressed frustration at their local Caldicott Guardian, the person with responsibility for protecting patient confidentiality and compliance with data protection regulations. When approached for guidance on the agreement within their Trust, the Caldicott Guardian had sent:

…a very bland reply saying well, 'if this is what the government want to do, and they've agreed, then we'll just do it'. [B02]

Conversely, B08, themselves a Caldicott Guardian, gave an opinion on the legality of the policy, and bemoaned lack of awareness among colleagues:

… I believe that [thumps table] what happened here, did not follow the processes which I certainly thought was there and then, as Caldicott Guardian, thought I was part of. [B08]

Participants also referred to tension between colleagues because of charging policies for overseas visitors. Two participants based in a hospital setting highlighted their negative experience of how Overseas Visitors Officers sought to recover charges from patients. The first clinician described a situation where a patient had been told minutes before surgery that they were ineligible and would have to pay for treatment. A second defined the role as having a 'degree of racial profiling' due to time constraints and how they 'suspect' or would 'be surprised if they had no role in data-sharing…' [B04], highlighting the perceived link between the two policies.

## DISCUSSION
### Key findings
We have shown that there is first, a varied understanding of the data-sharing agreement among healthcare professionals and volunteers who work closely with migrants, attributed to a lack of transparency in the development and dissemination of the policy by government. Second, we heard significant concerns about the impact of the data-sharing agreement on migrants' already compromised access to healthcare, and on the relationship between clinicians and patients which is founded on principles of trust and confidentiality.[65] Third, we identified worrying examples of how the data-sharing agreement could adversely affect quality and safety of patient care, through patients withholding information or clinicians potentially bypassing standard recordkeeping in an attempt to protect patient confidentiality.

The study resonated with previously reported barriers to migrants accessing health services, including perceived discrimination.[32 41] Participants referred to both healthcare charges and data-sharing policies in their narratives, suggesting that they are perceived to be closely related. Findings were influenced by the participant's clinical setting; sexual health providers emphasised confidentiality as a key feature, whereas a psychiatrist discussed how disclosing information was important for managing safety and 'risk'. This highlights an understanding that some data-sharing will always be required. However, there is a delicate balance, which must be communicated with patients to avoid creating any additional barriers to

accessing services. Overall, participants largely disagreed with patient data-sharing for immigration purposes and perceived the agreement as instilling fear of deportation, with some alluding to potential future data-sharing misuses with other government departments. This echoes Foucault's[66] description of pervasive power associated with surveillance and monitoring evident across society and intersecting with aspects of life including health and well-being. Some participants who knew less about the agreement were indecisive about the amendment made for 'serious' crime which may be due to a lack of understanding and insufficient time to consider their view. However, this also allowed the interviewer to explore the participants' immediate reactions, concerns and responses of sharing non-clinical patient data for immigration purposes.

### Strengths and limitations of the study
There were several strengths to this study. The interviews generated rich, contextual data with a reflexive process acknowledging the active role and influences of the researcher.[55] Coding and themes were discussed with authors experienced in healthcare for marginalised groups (HW) and public health research (HW and AW-S), providing additional methodological rigour. Furthermore, conducting the study as an academic (VP) provided the opportunity to ask challenging questions; for instance, exploring relationships with colleagues, an area overlooked in a priori themes. The main study limitation was sample size due to recruitment difficulties including the duration of the study and limited availability of healthcare providers. Further work is required to assess how common these experiences are by healthcare workers working in other clinical settings and geographical regions of the UK. It is likely that recruitment was influenced by the participants being aware of the study purpose which, in turn, may have influenced their responses. Additionally, the non-clinical background of VP will have influenced the analysis and interpretation of findings.

### Meaning of the study
Policy-makers should consider the implications of patient data-sharing agreements for individual and wider public health when monitoring their implementation. For instance, some vulnerable subgroups, such as pregnant women, were perceived to be negatively affected by the policy. However, these are only the cases that presented to healthcare and there will inevitably be those too afraid to access health services (including undocumented migrants) due to the policy. Second, the involvement of medical ethicists, independent advisors and Caldicott Guardians should be considered for future intragovernmental data-sharing arrangements that may impact healthcare access.[67] Participants suggested that stakeholder consultations would improve the understanding of governmental decision-making and would support the rebuilding of trust between government, the public

and NHS workforce. Finally, health policies should be considered in relation to the NHS principles as there may be consequences to clinical care delivered if these are undermined.

### Future work
Future research should explore the impact of data-sharing and other policies and contexts on migrant healthcare access and health. Further qualitative work should include all stakeholders including migrants, community organisations, health and social care providers and policy-makers, and be linked to quantitative research on service use, and larger surveys to capture concrete examples of issues in quality and safety. Conducting research in areas outside of London that have seen an influx of migration would be an interesting narrative to explore as at least 75% of participants interviewed in the study worked in the diverse capital.

### CONCLUSION
The study raises important and pertinent questions around the role of policies in influencing migrants' access, uptake and presentation to healthcare services. The study findings suggest that this data-sharing agreement created tension with the NHS principles. Although some case studies may not be directly linked to the data-sharing agreement, they indicated how blurring the lines between healthcare and immigration control can influence care received. As such, the formalisation, implementation and amendment of this policy could be defined as a significant example of where perceived lack of transparency can lead to suspicion, distrust and fear towards government by both NHS workers and patients and can potentially result in harm to both patient safety and the wider public health.

### List of recommendations
► Policy-makers monitoring the implementation of patient data-sharing agreements should consider how these may have implications to individual and the wider public health, including vulnerable individuals who may no longer present to healthcare services.
► Medical ethicists, independent advisors and Caldicott Guardians should be involved during the formalisation of future UK intragovernmental data-sharing agreements that may impact healthcare access.
► UK health policies should be considered in relation to the NHS principles in order to circumvent any potential consequences to the quality, and safety, of clinical care delivered.

**Acknowledgements** We would like to thank all the study participants for taking the time to share their views.

**Contributors** VP, HW, AW-S and IC-M: study concept and design; VP: conducted, transcribed and analysed all interviews and drafted the manuscript; HW and AW-S: supported with analysis of transcripts; HW, AW-S and IC-M: contributed to critical

advice and revisions of the manuscript. All authors read and approved the final manuscript.

**Funding** The research was supported by the National Institute for Health Research (NIHR) Biomedical Research Centre based at Imperial College Healthcare NHS Trust and Imperial College London.

**Disclaimer** The views expressed are those of the author(s) and not necessarily those of the NIHR, Public Health England or the Department of Health and Social Care.

**Competing interests** None declared.

**Patient consent for publication** Not required.

**Ethics approval** Ethical approval was obtained from Imperial College London (18IC4526) and Public Health England (R&D H088).

**Provenance and peer review** Not commissioned; externally peer reviewed.

**Data availability statement** Audio files were destroyed immediately after transcription with participants designated a non-identifiable code to ensure anonymity. Anonymised transcripts are stored in an encrypted and password-protected environment at Imperial College London. No additional data available.

**ORCID iD**
Vasiliki Papageorgiou http://orcid.org/0000-0002-2387-6780

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
