## [Reviewer comments · BMJ Open]

ARTICLE DETAILS

TITLE (PROVISIONAL)	Patient data sharing for immigration enforcement: a qualitative study of healthcare providers in England
AUTHORS	Papageorgiou, Vasiliki; Wharton-Smith, Alexandra; Campos-Matos, Ines; Ward, Helen

VERSION 1 – REVIEW

REVIEWER	Dr Claire Brolan Centre for Policy Futures
REVIEW RETURNED	23-Sep-2019

GENERAL COMMENTS	The subject matter under investigation in this study is very important and very topical, not only in the UK but also internationally. I have no doubt this manuscript would be of enormous interest to the journal's international readership. Certainly, this study has the promise and potential to make a significant and very important contribution to the literature. I can't help but reflect that this paper wholly intersects with, and adds a new twist to, the subject of 'health security'. However, if this manuscript is to be published then its content needs to be significantly strengthened. The manuscript interrogates sensitive issues around access to health services of vulnerable and marginalised populations, stigma, health justice, health and human rights, health ethics issues, e-health, migrant health, health practitioner ethics – all of which are politically charged in the UK, and elsewhere. The underlying messages and concerns identified in this manuscript are very, very important. However, I fear that in its current form, and without extensive review and rework by the authors, the paper is open to much criticism. It is for this reason that I have provided detailed comments on how I recommend the paper is reviewed and extensively revised, and greatly encourage the authors to rework this paper and take it to the proverbial 'next level'. This is because, as mentioned, the paper has much promise and potential to make a significant contribution to the literature. The main concerns that I have with the paper are as follows: 1. The weak introduction and background section. The authors need to stamp themselves as authorities in this field - please leverage the literature to do this, as well as the factual background.2. The Methods section – the Methods section is poor (sorry!) and because the Methods section is weak, this undermines the veracity of the results. The ethics around health practitioner
--

participant involvement (i.e. being interviewed as to their potentially lawful or unlawful actions) is also not acknowledged in this paper. How was this dealt with by the institutional human ethics team that gave approval to this study?

3. Paper limitations – a major limitation with this paper is that there is only 12 participants, many of whom are based in London, some of who have nothing substantive to add or comment because they have no knowledge on the policy. While the authors acknowledge the limited number of participants, they do not make the case for why despite this small sample number the research findings are nonetheless rigorous, valid, and generalizable. The exploratory yet credible nature of this study needs to be emphasised (not that it is a postgraduate studies piece of research).

Again, I wish to reiterate to the authors that the detailed feedback and critique I give is in the interests of having the paper strengthened for publication. This paper could make a very important contribution to the literature, on a number of fronts, but it is not there yet. Again, I commend the authors for identifying and pursuing this piece of important exploratory, qualitative research. However, in order for this paper to be published, all co-authors must own this scientific study and support the substantial revision of the manuscript.

Introduction

General comment –

I suggest that the authors insert a Chronology of Events relating to ‘the agreement’ into the Introduction (i.e Figure 1). This would be of enormous assistance for the reader.

The authors cannot assume the reader has any background knowledge on the UK context (especially the journal’s international readership). The authors, in the Introduction, really need to take the reader on the journey – so there needs to be more specificity in the Introduction. E.g.:

Paragraph 1

- ‘However concerns were raised about the potential impact...’ – By who? A single individual or agency? Multiple agencies, such as... ? Please be specific.
- ‘Following these objections, an amendment was made to restrict requests...’ – Who made the amendment? Please be specific, and when was the amendment made?
- Please explain to the journal’s readership what is the new General Data Protection Regulation? Who introduced this, what is the object of this regulation? (only one sentence is needed)
- ‘The new agreement has yet to be published however...’ again, please be specific. It is a little unclear as to which ‘agreement’ is meant here. And, who then issued the revision plan?

Paragraph 2

- How is migration a ‘core determinant of health and wellbeing’. Indeed, how? What does the literature say? What are the ‘various factors’ that have been shown to influence the health seeking behaviour and access to healthcare services of both documented and undocumented migrants? Are these examples specifically from the UK context? Elsewhere? Again, please take the reader on the journey with you. Be very mindful of the narrative and the flow of the Introduction. Please ensure all statements of fact are grounded in the literature (and robust literature).

	Paragraph 3  • ‘To date, most UK studies...’ - What is this literature? Describing this literature and its focus is important to contextualise this study, and to highlight its significance. The authors need to present themselves as ‘experts’ on the study subject matter. Thus, grounding the paper in the UK (and broader) literature is very important. Paragraph 4  • ‘Concerns have been reported about the collection of individuals’ migration status in healthcare systems globally, particularly regarding the potential misuse of health data and critics emphasise that patient confidentiality and trust are incompatible with the data sharing agreement.’  o This opening sentence should be broken down into 2 sentences. Is the concern around the potential misuse of health data or is the real concern around the misuse of patient identification data (i.e. migration status, address etc). Is it both? • The second sentence in Paragraph 4 may be true, but there are a number of potentially controversial assertions made in this sentence, and the authors need to ground these assertions in and with strong reference material. The alleged ‘hostile environment’ toward migration that the authors contend is allegedly promoted by the UK Government, should this not also be nuanced in the broader international context, wherein many high-income countries are tightening their immigration/border control. This is not only a UK phenomenon. • The comment ‘in the application of immigration policy to other areas...’ is, again not authoritative. Its ‘meaning’ is unclear. The authors need to be more specific. You are the research specialists on this topic, and must present yourselves as the specialist knowledge-holders. This is particularly imperative given the potentially sensitive nature of the cross-cutting issue areas that this research explores. • ‘This sits uncomfortably with the increased onus on public bodies to regulate health and care data’ – should this be health care data? And, who is placing this onus on public bodies? Which public bodies (ensure citation of the literature? Again please be more specific and authoritative. • ‘Additionally, there is limited research on the potential...’ is this limited research in the UK context or more broadly? If the research is ‘limited’ then this implies there is some research on the potential conflicts. What is this research, and again, is this in the UK context? • ‘Therefore, it is important to understand...’ this should be the beginning of a new paragraph. • ‘Therefore, it is important to understand the perceptions of the agreement by healthcare workers...’ – what agreement, please be more specific. And, perceptions of which healthcare workers in particular (and why these healthcare workers), and where (do you mean UK health-care workers; yes, of course you do, but please be more specific – take the reader on the journey). • Has this research been conducted (i.e. specifically exploring the views of healthcare workers) before? In the UK or elsewhere? The authors need to emphasise why this study is an important and new contribution to the literature. Methods The Methods section needs to begin with a statement around the study aims and objectives. The ‘study design’ sub-section is
--	--

	actually not 'a study design' and, respectfully, needs to be wholly revised. This is a particular weakness of the paper. If the Methods section is not strong, then this puts doubts in the reviewer's mind as to the veracity of the findings. Please revise. If the study has received institutional human research ethics clearance approval then there is no need to include the sentence 'The study formed the thesis...', so please remove. Also, the human research ethics clearance approval statement should be cut/paste to become the final sentence of the Methods. Sampling sub-section Again, regrettably, the detail in the sampling sub-section is not explicit enough. E.g. how were the participants recruited exactly? Was there snowball sampling? Verbal or written consent? Who are the participants (healthcare workers – but what do the authors mean???), were the participants in urban or rural areas? Were they recruited from locations that are known to have high-levels of migrant populations? How did the research team obtain consent/authorisation of the healthcare authorities to interview the healthcare workers? Were the potential participants sent a Participant Information Sheet and/or consent form beforehand? Data collection Please remove 'female, postgraduate researcher' and instead state 'A member of the research team conducted...' Suddenly there is mention of the 'constructionist concept' this comes out of left field. There needs to be more critical engagement and explanation with this concept, and why it was utilised and integrated by the research team. And, what particular constructionist approach was followed? The theory here is weak. There is detail in the data collection that should actually be part of the data analysis. Data analysis The sudden introduction of grounded theory makes no sense. Again, this relates to the study design, which needs to be completely revised. Please explain the process of inter-rata reliability; I'm not convinced by the statement 'HW and AWS provided consensus on coding'. Please remove the sub-section on patient and public involvement. Results What was the average interview time length? Strengths and limitations of the study – A concise and express 'List of Recommendations' would be very good. These could also be placed in a Box in summary form.
--	--

REVIEWER	Megan A. Carney University of Arizona, USA
REVIEW RETURNED	24-Sep-2019

GENERAL COMMENTS	The authors of this paper examine the perceptions and experiences of healthcare providers in response to recent legislation that has infringed on the privacy of patients. Specifically, this legislation seeks to identify "immigrant offenders" for the means of detaining and deporting them. Consistent with the
--

	biopolitics of immigration and citizenship (Gonzales and Chavez 2012), this law allows for unethical practices in the surveillance and policing of patients. It not only infringes on the rights of noncitizens by heightening their visibility to the state, but it arguably infringes on the rights of citizens as well. In short, it undermines health as a human right and prioritizes state sovereignty practices over population health and wellbeing. This particular study is especially significant in that it reveals the implications of this specific and similar types of legislation in shaping healthcare seeking behaviors of vulnerable populations. Such changes in healthcare seeking behaviors portend real consequences for public health. For the most part, the authors outline a robust qualitative study. The methods are clearly stated and the background literature is adequate (but could be more robust) for introducing the specific focus of this study. However, the authors could be more detailed in their presentation of results. It is not obvious that they achieved data saturation as they claimed per the qualitative findings that are highlighted. Could they expand on these findings, perhaps provide more examples from interviews and give more context to the findings? This would help to justify the discussion and conclusions of this study.
--	--

VERSION 1 – AUTHOR RESPONSE

Reviewer 1 general comment

The subject matter under investigation in this study is very important and very topical, not only in the UK but also internationally. I have no doubt this manuscript would be of enormous interest to the journal's international readership. Certainly, this study has the promise and potential to make a significant and very important contribution to the literature. I can't help but reflect that this paper wholly intersects with, and adds a new twist to, the subject of 'health security'.

However, if this manuscript is to be published then its content needs to be significantly strengthened. The manuscript interrogates sensitive issues around access to health services of vulnerable and marginalised populations, stigma, health justice, health and human rights, health ethics issues, e-health, migrant health, health practitioner ethics – all of which are politically charged in the UK, and elsewhere. The underlying messages and concerns identified in this manuscript are very, very important. However, I fear that in its current form, and without extensive review and rework by the authors, the paper is open to much criticism.

It is for this reason that I have provided detailed comments on how I recommend the paper is reviewed and extensively revised, and greatly encourage the authors to rework this paper and take it to the proverbial 'next level'. This is because, as mentioned, the paper has much promise and potential to make a significant contribution to the literature.

The main concerns that I have with the paper are as follows:

1. The weak introduction and background section. The authors need to stamp themselves as authorities in this field - please leverage the literature to do this, as well as the factual background.
2. The Methods section – the Methods section is poor (sorry!) and because the Methods section is weak, this undermines the veracity of the results. The ethics around health practitioner participant involvement (i.e. being interviewed as to their potentially lawful or unlawful actions) is also not acknowledged in this paper. How was this dealt with by the institutional human ethics team that gave approval to this study?
3. Paper limitations – a major limitation with this paper is that there is only 12 participants, many of whom are based in London, some of who have nothing substantive to add or comment because they have no knowledge on the policy. While the authors acknowledge the limited number of participants, they do not make the case for why despite this small sample number the research findings are nonetheless rigorous, valid, and generalizable. The exploratory yet credible nature of this study needs to be emphasised (not that it is a postgraduate studies piece of research).

Again, I wish to reiterate to the authors that the detailed feedback and critique I give is in the interests of having the paper strengthened for publication. This paper could make a very important contribution to the literature, on a number of fronts, but it is not there yet. Again, I commend the authors for identifying and pursuing this piece of important exploratory, qualitative research. However, in order for this paper to be published, all co-authors must own this scientific study and support the substantial revision of the manuscript.

Author's response and revisions:

We thank Reviewer 1 for their supportive, and detailed comments on our work. We are delighted that the work is of interest and we are keen to ensure that the manuscript reflects the high quality and rigorous approach taken. In terms of the specific points raised above (points 1 to 3), we would like to add:

1. **Introduction** – we have made significant changes to the manuscript in the revised version and have added some more recently published, significant literature.
2. **Methods** – we appreciate that there were some omissions in the original methods section that would have been useful for clarity and so we have added further detail in terms of the theoretical underpinning of the study and ethical approval received. Regarding the ethical agreement sought and received, the study was approved through Imperial College London, rather than NHS ethics. As a result, no participants were recruited directly through their employer (i.e. NHS Trust) and interviews could not be conducted at any NHS Trust sites, so were instead conducted at the University campus or by telephone.
3. **Limitations** – we have added more details to the manuscript (namely in the Discussion) to reflect on the reviewer's comments in terms of sample size and credibility of the study. We feel that discovering that most interviewees were not aware of the agreement was a finding in itself; this suggests a lack of communication by government to healthcare providers but also how healthcare providers learn about health policies (e.g. through colleagues or professional bodies). This also allowed us to explore the immediate reaction of interviewees to the agreement which meant we could use more probing questions.

The table below outlines the specific comments made by Reviewer 1 and our responses and revisions made. Once again, we'd like to thank Reviewer 1 for their thorough comments.

Reviewer 1 comment	Authors' response and revisions
Introduction	
I suggest that the authors insert a Chronology of Events relating to 'the agreement' into the Introduction (i.e Figure 1). This would be of enormous assistance for the reader. The authors cannot assume the reader has any background knowledge on the UK context (especially the journal's international readership). The authors, in the Introduction, really need to take the reader on the journey – so there needs to be more specificity in the Introduction.	We agree that a visual representation of the chronological order leading to the formalisation and suspension of the patient data sharing agreement would be helpful for the reader and so have created a new Figure 1.
Paragraph 1  1. 'However concerns were raised about the potential impact...' – By who? A single individual or agency? Multiple agencies, such as... ? Please be specific. 2. 'Following these objections, an amendment was made to restrict requests...' – Who made the amendment? Please be specific, and when was the amendment made? 3. Please explain to the journal's readership what is the new General Data Protection Regulation? Who introduced this, what is the object of this regulation? (only one sentence is needed) 4. 'The new agreement has yet to be published however...' again, please be specific. It is a little unclear as to which 'agreement' is meant here. And, who then issued the revision plan? 	(see page 4)  1. We have added further detail regarding the concerns of the agreement by the public health community and advocacy groups. 2. We have added details regarding the amendment to the UK Home Office/NHS Digital data sharing agreement referring to the House of Commons debate of the Data Protection Bill in May 2018 and include a new reference [9]. 9. Data Protection Bill [Lords]. House of Commons Hansard. London, 2018.  3. We include further detail regarding the EU General Data Protection Regulation and include a new reference [12]. 12. EU General Data Protection Regulation (GDPR). Regulation (EU) 2016/679 of the European Parliament and of the Council of 27 April 2016 on the protection of natural persons with regard to the processing of personal data and on the free movement of such data, and repealing Directive 95/46/EC (General Data Protection Regulation). OJ 2016 L 119/1.  4. We add further clarity of what is meant by 'agreement' and the revision plan published on the NHS Digital website.
Paragraph 2  1. How is migration a 'core determinant of health and wellbeing'. Indeed, how? What does the literature say? 2. What are the 'various factors' that have been shown to influence the health seeking behaviour and access to healthcare services of both documented and undocumented migrants? Are these examples specifically from the UK context? Elsewhere? 	(see pages 4-5)  1. We have added further details to this paragraph – significantly, the reasons why someone may migrate and the influencers to their health that they may experience before, during or after the migration journey. We include the addition of 4 new references [17-20].

Again, please take the reader on the journey with you. Be very mindful of the narrative and the flow of the Introduction. Please ensure all statements of fact are grounded in the literature (and robust literature).	17. Rechel B, Mladovsky P, Ingleby D, et al. Migration and health in an increasingly diverse Europe. The Lancet 2013;381(9873):1235-45. doi: 10.1016/S01406736(12)62086-8 18. Gushulak BD, MacPherson DW. The basic principles of migration health: Population mobility and gaps in disease prevalence. Emerging Themes in Epidemiology 2006;3(1):3. doi: 10.1186/1742-7622-33 19. Robertshaw L, Dhesi S, Jones LL. Challenges and facilitators for health professionals providing primary healthcare for refugees and asylum seekers in high-income countries: a systematic review and thematic synthesis of qualitative research. BMJ Open 2017;7(8):e015981. doi: 10.1136/bmjopen-2017015981 20. Zimmerman C, Kiss L, Hossain M. Migration and Health: A Framework for 21st Century Policy-Making. PLOS Medicine 2011;8(5):e1001034. doi: 10.1371/journal.pmed.1001034 2. We refer to these factors in the sentence, "...including cultural ideas and expectations, socio-economic conditions, geographical and institutional factors" however we have added that this remains context-specific (the studies referenced are from a UK, EU and global perspective).
Paragraph 3 'To date, most UK studies...' - What is this literature? Describing this literature and its focus is important to contextualise this study,	(see page 5) We have provided some more information regarding the type of studies that have been
and to highlight its significance. The authors need to present themselves as 'experts' on the study subject matter. Thus, grounding the paper in the UK (and broader) literature is very important.	published in the UK to date being predominantly of qualitative or observational in nature. We include a new reference [41] and have added further examples of qualitative/observational studies in the final sentence – some of these studies were previously referred to in the article (apart from reference 41). 41. Stagg HR, Jones J, Bickler G, et al. Poor uptake of primary healthcare registration among recent entrants to the UK: a retrospective cohort study. BMJ Open 2012;2(4):e001453. doi: 10.1136/bmjopen-2012001453

Paragraph 4  1. 'Concerns have been reported about the collection of individuals' migration status in healthcare systems globally, particularly regarding the potential misuse of health data and critics emphasise that patient confidentiality and trust are incompatible with the data sharing agreement.' This opening sentence should be broken down into 2 sentences. Is the concern around the potential misuse of health data or is the real concern around the misuse of patient identification data (i.e. migration status, address etc). Is it both? 2. The second sentence in Paragraph 4 may be true, but there are a number of potentially controversial assertions made in this sentence, and the authors need to ground these assertions in and with strong reference material. The alleged 'hostile environment' toward migration that the authors contend is allegedly promoted by the UK Government, should this not also be nuanced in the broader international context, wherein many high-income countries are tightening their immigration/border control. This is not only a UK phenomenon. 3. The comment 'in the application of immigration policy to other areas...' is, again not authoritative. Its 'meaning' is unclear. The authors need to be more specific. You are the research specialists on this topic, and must present yourselves as the specialist knowledgeable holders. This is particularly imperative given the potentially sensitive nature of the cross-cutting issue areas that this research explores. 4. 'This sits uncomfortably with the increased onus on public bodies to regulate health and care data' – should this be health care data? And, who is placing this onus on public bodies? Which public bodies (ensure citation of the literature? Again please be more specific and authoritative. 5. 'Additionally, there is limited research on the potential...' is this limited research in the UK context or more broadly? If the research is 'limited' then this implies there is some research on the potential conflicts. What is this research, and again, is this in the UK context? 	(see page 5)  1. We have made the suggested changes and have added the explanation that both types of data misuse are concerning and have made it clearer regarding the type of sensitive data shared as part of the UK Home Office/NHS Digital agreement. We have added an additional reference [45] of a recently published study exploring the NHS charging regulations in the UK which supports this statement. 45. Reynolds JMK, Mitchell C. 'Inglan is a bitch': hostile NHS charging regulations contravene the ethical principles of the medical profession. Journal of Medical Ethics 2019;45(8):497-503. doi: 10.1136/medethics2019-105419 2. Agreed – we have added 2 new references [42, 43] referring to EU / US immigration policies and impact to health outcomes to the start of paragraph 4. 42. Khullar D, Chokshi DA. Challenges for immigrant health in the USA-the road to crisis. The Lancet 2019;393(10186):2168-74. doi: 10.1016/S01406736(19)30035-2 43. Winters M, Rechel B, de Jong L, et al. A systematic review on the use of healthcare services by undocumented migrants in Europe. BMC Health Services Research 2018;18(1):30. doi: 10.1186/s12913-018-2838-y 3. We have added detail regarding the Immigration Act 2016 and its implications for support given to rejected asylum seekers and two additional references [48, 49]. 48. Home Office, UK Visas and Immigration. Immigration Act: overview 2016 [updated 12 July 2016. Available from: https://www.gov.uk/government/publications/immigration-bill-2015-overarching-documents/immigration-bill201516-overview-factsheet accessed 18 October 2019. 49. Home Office. Immigration Act 2016: Factsheet – Support for certain categories of migrants (Section 66) 2016 [updated July 2016. Available from: https://assets.publishing.service.gov.uk/government/uploads/system/uploads/attachment_data/file/537248/Immigration Act - Part 5 - Support for Certain Categories of migrants.pdf. 4. Here, we describe the network of UK Caldicott Guardian Council
---	--

6. 'Therefore, it is important to understand...' this should be the beginning of a new paragraph. 7. 'Therefore, it is important to understand the perceptions of the agreement by healthcare workers...' – what agreement, please be more specific. And, perceptions of which healthcare workers in particular (and why these healthcare workers), and where (do you mean UK health-care workers; yes, of course you do, but please be more specific – take the reader on the journey). 8. Has this research been conducted (i.e. specifically exploring the views of healthcare workers) before? In the UK or elsewhere? The authors need to emphasise why this study is an important and new contribution to the literature.	(organisational body) and Information Commissioner's Office (public body) responsible for regulating the implementation of GDPR. Both health and social care data are protected and so we have added the word 'social' to make this clearer. 5. We have clarified the context however the limited, robust research is outlined in the sentence of the Introduction ("the study aims to contribute..."). We have added an additional reference here [52] to support this statement. 52. Saadi A, Ahmed S, Katz MH. Making a Case for Sanctuary Hospitals. JAMA 2017;318(21):2079-80. doi: 10.1001/jama.2017.15714 6. Agreed and we have made a new paragraph. 7. We have added in UK Home Office/NHS Digital to clarify the data sharing agreement. We believe the views of all healthcare workers are important as they will each meet migrant patients for several different health reasons. To our knowledge, research has not been conducted before with healthcare workers exploring this specific agreement (UK Home Office/NHS Digital) however, some editorials, opinion-based pieces and NGO reports have been published which are referred to in the penultimate sentence. We have added a sentence to clarify this and refer to these published findings in this sentence, including a patient case study published in July 2019 [45] – the study was published following our initial submission to BMJ Open. 45. Reynolds JMK, Mitchell C. 'Inglan is a bitch': hostile NHS charging regulations contravene the ethical principles of the medical profession. Journal of Medical Ethics 2019;45(8):497-503. doi: 10.1136/medethics2019-105419
--	---

Methods	
The Methods section needs to begin with a statement around the study aims and objectives. The 'study design' sub-section is actually not 'a study design' and, respectfully, needs to be wholly revised. This is a particular weakness of the paper. If the Methods section is not strong, then this puts doubts in the reviewer's mind as to the veracity of the findings. Please revise.	The study aim is given in the penultimate sentence of the introduction (see page 6). We believe this sub-section is describing the 'study design' as it describes the approach (qualitative) and conceptual framework (using NHS principles). We have added some further information regarding the theoretical underpinnings.

	We add an additional reference [56] to explain the role of recognising an interviewer’s own bias in an exploratory, qualitative study. 56. Lincoln YS, Guba EG. Naturalistic Inquiry: SAGE Publications 1985:363-364.
If the study has received institutional human research ethics clearance approval then there is no need to include the sentence ‘The study formed the thesis...’, so please remove. Also, the human research ethics clearance approval statement should be cut/paste to become the final sentence of the Methods.	We have moved the ethical approval statement to the end of the Methods section (see page 8) as suggested, under a new “ethical approval” sub-section.
Sampling sub-section: Again, regrettably, the detail in the sampling sub-section is not explicit enough.  1. E.g. how were the participants recruited exactly? 2. Was there snowball sampling? 3. Verbal or written consent? 4. Who are the participants (healthcare workers – but what do the authors mean???) 5. were the participants in urban or rural areas? 6. Were they recruited from locations that are known to have high-levels of migrant populations? 7. How did the research team obtain consent/authorisation of the healthcare authorities to interview the healthcare workers? 	(see page 7)  1. Outlined in second paragraph (personal contacts, social media etc.). Recruitment was stopped at 12 participants due to time constraints. 2. No 3. Yes, this is detailed in final sentence of “data collection” sub-section which we have now moved to “ethical approval” sub-section. 4. We have added some more information regarding the professional backgrounds of the participants (medical doctors, physiotherapist and voluntary worker). 5. Urban areas in England – we have added this to first sentence.
8. Were the potential participants sent a Participant Information Sheet and/or consent form beforehand?	 6. We only collected broad geographical location data before interviews so are unable to comment on this. We have added this for clarification into the second paragraph. 7. As interviews were not conducted on NHS Trust sites and participants were not directly recruited through their place of work, we did not need to obtain consent from NHS HRA ethics. 8. Yes, we have added this to the final sentence in the “recruitment and sampling” sub-section for clarity.

Data collection 1. Please remove 'female, postgraduate researcher' and instead state 'A member of the research team conducted...' 2. Suddenly there is mention of the 'constructionist concept' this comes out of left field. There needs to be more critical engagement and explanation with this concept, and why it was utilised and integrated by the research team. And, what particular constructionist approach was followed? The theory here is weak. There is detail in the data collection that should actually be part of the data analysis.	(see pages 7-8) 1. Have removed. 2. Have added further detail on constructionist approach into 'study design' subsection.
Data analysis 1. The sudden introduction of grounded theory makes no sense. Again, this relates to the study design, which needs to be completely revised. 2. Please explain the process of inter-rata reliability; I'm not convinced by the statement 'HW and AWS provided consensus on coding'.	(see page 8) 1. The study design was not guided by grounded theory however, we used the feature of 'constant-comparison' to analyse the transcripts. 2. HW and AWS checked the coding framework of the first three transcripts coded by VP and were approached by VP to check any interpretations made on subsequent transcripts. We have added this statement to the end of the first paragraph.
Please remove the sub-section on patient and public involvement.	We have included this statement as it is required by the journal: https://authors.bmj.com/policies/patientpublic-partnership/
Results	
What was the average interview time length?	We have added this information (average: 40 minutes) – see page 9.
Discussion	
Strengths and limitations of the study – A concise and express 'List of Recommendations' would be very good. These could also be placed in a Box in summary form.	We have added further detail regarding the background of two of the authors involved in data analysis (HW, AWS) as well as limitations regarding recruitment and the generalisability of findings. We have summarised the recommendations listed in the 'meaning of the study' subsection in 3 bullet points after the conclusion.

Reviewer 2 comment:

The authors of this paper examine the perceptions and experiences of healthcare providers in response to recent legislation that has infringed on the privacy of patients. Specifically, this legislation seeks to identify "immigrant offenders" for the means of detaining and deporting them. Consistent with the biopolitics of immigration and citizenship (Gonzales and Chavez 2012), this law allows for unethical practices in the surveillance and policing of patients. It not only infringes on the rights of noncitizens by heightening their visibility to the state, but it arguably infringes on the rights of citizens

as well. In short, it undermines health as a human right and prioritizes state sovereignty practices over population health and wellbeing.

This particular study is especially significant in that it reveals the implications of this specific and similar types of legislation in shaping healthcare seeking behaviors of vulnerable populations. Such changes in healthcare seeking behaviors portend real consequences for public health.

For the most part, the authors outline a robust qualitative study. The methods are clearly stated and the background literature is adequate (but could be more robust) for introducing the specific focus of this study. However, the authors could be more detailed in their presentation of results. It is not obvious that they achieved data saturation as they claimed per the qualitative findings that are highlighted. Could they expand on these findings, perhaps provide more examples from interviews and give more context to the findings? This would help to justify the discussion and conclusions of this study.

Author's response and revisions:

We would like to thank Reviewer 2 for their supportive comments and feedback. We agree that the introduction required some more robust literature and so have added further information for clarity (see table above).

We stopped recruitment at 12 participants due to the time constraints of the study and so have removed any detail of theoretical saturation being met as we are unable to say that this is the case. However, we agree that some further detail should be given regarding some of the results presented and so we have added some more information under the heading of 'confidentiality' as this was a key finding from our study and to justify its description in the Discussion

We'd like to thank Reviewer 2 again for their comments.